# Tree-Guided MCMC Inference for Normalized Random Measure Mixture Models

**Juho Lee and Seungjin Choi**
Department of Computer Science and Engineering
Pohang University of Science and Technology
77 Cheongam-ro, Nam-gu, Pohang 37673, Korea
{stonecold,seungjin}@postech.ac.kr

## Abstract

Normalized random measures (NRMs) provide a broad class of discrete random measures that are often used as priors for Bayesian nonparametric models. Dirichlet process is a well-known example of NRMs. Most of posterior inference methods for NRM mixture models rely on MCMC methods since they are easy to implement and their convergence is well studied. However, MCMC often suffers from slow convergence when the acceptance rate is low. Tree-based inference is an alternative deterministic posterior inference method, where Bayesian hierarchical clustering (BHC) or incremental Bayesian hierarchical clustering (IBHC) have been developed for DP or NRM mixture (NRMM) models, respectively. Although IBHC is a promising method for posterior inference for NRMM models due to its efficiency and applicability to online inference, its convergence is not guaranteed since it uses heuristics that simply selects the best solution after multiple trials are made. In this paper, we present a hybrid inference algorithm for NRMM models, which combines the merits of both MCMC and IBHC. Trees built by IBHC outlines partitions of data, which guides Metropolis-Hastings procedure to employ appropriate proposals. Inheriting the nature of MCMC, our tree-guided MCMC (tgMCMC) is guaranteed to converge, and enjoys the fast convergence thanks to the effective proposals guided by trees. Experiments on both synthetic and real-world datasets demonstrate the benefit of our method.

## 1 Introduction

Normalized random measures (NRMs) form a broad class of discrete random measures, including Dirichlet proccess (DP) [1] normalized inverse Gaussian process [2], and normalized generalized Gamma process [3, 4]. NRM mixture (NRMM) model [5] is a representative example where NRM is used as a prior for mixture models. Recently NRMs were extended to dependent NRMs (DNRMs) [6, 7] to model data where exchangeability fails. The posterior analysis for NRM mixture (NRMM) models has been developed [8, 9], yielding simple MCMC methods [10]. As in DP mixture (DPM) models [11], there are two paradigms in the MCMC algorithms for NRMM models: (1) marginal samplers and (2) slice samplers. The marginal samplers simulate the posterior distributions of partitions and cluster parameters given data (or just partitions given data provided that conjugate priors are assumed) by marginalizing out the random measures. The marginal samplers include the Gibbs sampler [10], and the split-merge sampler [12], although it was not formally extended to NRMM models. The slice sampler [13] maintains random measures and explicitly samples the weights and atoms of the random measures. The term "slice" comes from the auxiliary slice variables used to control the number of atoms to be used. The slice sampler is known to mix faster than the marginal Gibbs sampler when applied to complicated DNRM mixture models where the evaluation of marginal distribution is costly [7].

The main drawback of MCMC methods for NRMM models is their poor scalability, due to the nature of MCMC methods. Moreover, since the marginal Gibbs sampler and slice sampler iteratively sample the cluster assignment variable for a single data point at a time, they easily get stuck in local optima. Split-merge sampler may resolve the local optima problem to some extent, but is still problematic for large-scale datasets since the samples proposed by split or merge procedures are rarely accepted. Recently, a deterministic alternative to MCMC algorithms for NRM (or DNRM) mixture models were proposed [14], extending Bayesian hierarchical clustering (BHC) [15] which was developed as a tree-based inference for DP mixture models. The algorithm, referred to as incremental BHC (IBHC) [14] builds binary trees that reflects the hierarchical cluster structures of datasets by evaluating the approximate marginal likelihood of NRMM models, and is well suited for the incremental inferences for large-scale or streaming datasets. The key idea of IBHC is to consider only exponentially many posterior samples (which are represented as binary trees), instead of drawing indefinite number of samples as in MCMC methods. However, IBHC depends on the heuristics that chooses the best trees after the multiple trials, and thus is not guaranteed to converge to the true posterior distributions.

In this paper, we propose a novel MCMC algorithm that elegantly combines IBHC and MCMC methods for NRMM models. Our algorithm, called the tree-guided MCMC, utilizes the trees built from IBHC to proposes a good quality posterior samples efficiently. The trees contain useful information such as dissimilarities between clusters, so the errors in cluster assignments may be detected and corrected with less efforts. Moreover, designed as a MCMC methods, our algorithm is guaranteed to converge to the true posterior, which was not possible for IBHC. We demonstrate the efficiency and accuracy of our algorithm by comparing it to existing MCMC algorithms.

## 2 Background

Throughout this paper we use the following notations. Denote by $[n] = \{1, \ldots, n\}$ a set of indices and by $X = \{x_i \mid i \in [n]\}$ a dataset. A partition $\Pi_{[n]}$ of $[n]$ is a set of disjoint nonempty subsets of $[n]$ whose union is $[n]$. Cluster $c$ is an entry of $\Pi_{[n]}$, i.e., $c \in \Pi_{[n]}$. Data points in cluster $c$ is denoted by $X_c = \{x_i \mid i \in c\}$ for $c \in \Pi_n$. For the sake of simplicity, we often use $i$ to represent a singleton $\{i\}$ for $i \in [n]$. In this section, we briefly review NRMM models, existing posterior inference methods such as MCMC and IBHC.

### 2.1 Normalized random measure mixture models

Let $\mu$ be a homogeneous completely random measure (CRM) on measure space $(\Theta, \mathcal{F})$ with Lévy intensity $\rho$ and base measure $H$, written as $\mu \sim \mathrm{CRM}(\rho H)$. We also assume that,

$$\int_0^\infty \rho(dw) = \infty, \quad \int_0^\infty (1 - e^{-w})\rho(dw) < \infty, \tag{1}$$

so that $\mu$ has infinitely many atoms and the total mass $\mu(\Theta)$ is finite: $\mu = \sum_{j=1}^\infty w_j \delta_{\theta_j^*}$, $\mu(\Theta) = \sum_{j=1}^\infty w_j < \infty$. A NRM is then formed by normalizing $\mu$ by its total mass $\mu(\Theta)$. For each index $i \in [n]$, we draw the corresponding atoms from NRM, $\theta_i | \mu \sim \mu/\mu(\Theta)$. Since $\mu$ is discrete, the set $\{\theta_i | i \in [n]\}$ naturally form a partition of $[n]$ with respect to the assigned atoms. We write the partition as a set of sets $\Pi_{[n]}$ whose elements are non-empty and non-overlapping subsets of $[n]$, and the union of the elements is $[n]$. We index the elements (clusters) of $\Pi_{[n]}$ with the symbol $c$, and denote the unique atom assigned to $c$ as $\theta_c$. Summarizing the set $\{\theta_i | i \in [n]\}$ as $(\Pi_{[n]}, \{\theta_c | c \in \Pi_{[n]}\})$, the posterior random measure is written as follows:

**Theorem 1.** *([9]) Let $(\Pi_{[n]}, \{\theta_c | c \in \Pi_{[n]}\})$ be samples drawn from $\mu/\mu(\Theta)$ where $\mu \sim \mathrm{CRM}(\rho H)$. With an auxiliary variable $u \sim \mathrm{Gamma}(n, \mu(\Theta))$, the posterior random measure is written as*

$$\mu | u + \sum_{c \in \Pi_{[n]}} w_c \delta_{\theta_c}, \tag{2}$$

*where*

$$\rho_u(dw) := e^{-uw}\rho(dw), \quad \mu | u \sim \mathrm{CRM}(\rho_u H), \quad P(dw_c) \propto w_c^{|c|}\rho_u(dw_c). \tag{3}$$

*Moreover, the marginal distribution is written as*

$$P(\Pi_{[n]}, \{d\theta_c | c \in \Pi_{[n]}\}, du) = \frac{u^{n-1} e^{-\psi_\rho(u)} du}{\Gamma(n)} \prod_{c \in \Pi_{[n]}} \kappa_\rho(|c|, u) H(d\theta_c), \qquad (4)$$

*where*

$$\psi_\rho(u) := \int_0^\infty (1 - e^{-uw}) \rho(dw), \quad \kappa_\rho(|c|, u) := \int_0^\infty w^{|c|} \rho_u(dw). \qquad (5)$$

Using (4), the predictive distribution for the novel atom $\theta$ is written as

$$P(d\theta | \{\theta_i\}, u) \propto \kappa_\rho(1, u) H(d\theta) + \sum_{c \in \Pi_{[n]}} \frac{\kappa_\rho(|c| + 1, u)}{\kappa_\rho(|c|, u)} \delta_{\theta_c}(d\theta). \qquad (6)$$

The most general CRM may be used is the generalized Gamma [3], with Lévy intensity $\rho(dw) = \frac{\alpha\sigma}{\Gamma(1-\sigma)} w^{-\sigma-1} e^{-w} dw$. In NRMM models, the observed dataset $X$ is assusmed to be generated from a likelihood $P(dx|\theta)$ with parameters $\{\theta_i\}$ drawn from NRM. We focus on the conjugate case where $H$ is conjugate to $P(dx|\theta)$, so that the integral $P(dX_c) := \int_\Theta H(d\theta) \prod_{i \in c} P(dx_i|\theta)$ is tractable.

## 2.2 MCMC Inference for NRMM models

The goal of posterior inference for NRMM models is to compute the posterior $P(\Pi_{[n]}, \{d\theta_c\}, du | X)$ with the marginal likelihood $P(dX)$.

**Marginal Gibbs Sampler**: marginal Gibbs sampler is basesd on the predictive distribution (6). At each iteration, cluster assignments for each data point is sampled, where $x_i$ may join an existing cluster $c$ with probability proportional to $\frac{\kappa_\rho(|c|+1,u)}{\kappa_\rho(|c|,u)} P(dx_i|X_c)$, or create a novel cluster with probability proportional to $\kappa_\rho(1, u) P(dx_i)$.

**Slice sampler**: instead of marginalizing out $\mu$, slice sampler explicitly sample the atoms and weights $\{w_j, \theta_j^*\}$ of $\mu$. Since maintaining infinitely many atoms is infeasible, slice variables $\{s_i\}$ are introduced for each data point, and atoms with masses larger than a threshold (usually set as $\min_{i \in [n]} s_i$) are kept and remaining atoms are added on the fly as the threshold changes. At each iteration, $x_i$ is assigned to the $j$th atom with probability $\mathbb{1}[s_i < w_j] P(dx_i|\theta_j^*)$.

**Split-merge sampler**: both marginal Gibbs and slice sampler alter a single cluster assignment at a time, so are prone to the local optima. Split-merge sampler, originally developed for DPM, is a marginal sampler that is based on (6). At each iteration, instead of changing individual cluster assignments, split-merge sampler splits or merges clusters to propose a new partition. The split or merged partition is proposed by a procedure called the restricted Gibbs sampling, which is Gibbs sampling restricted to the clusters to split or merge. The proposed partitions are accepted or rejected according to Metropolis-Hastings schemes. Split-merge samplers are reported to mix better than marginal Gibbs sampler.

## 2.3 IBHC Inference for NRMM models

Bayesian hierarchical clustering (BHC, [15]) is a probabilistic model-based agglomerative clustering, where the marginal likelihood of DPM is evaluated to measure the dissimilarity between nodes. Like the traditional agglomerative clustering algorithms, BHC repeatedly merges the pair of nodes with the smallest dissimilarities, and builds binary trees embedding the hierarchical cluster structure of datasets. BHC defines the generative probability of binary trees which is maximized during the construction of the tree, and the generative probability provides a lower bound on the marginal likelihood of DPM. For this reason, BHC is considered to be a posterior inference algorithm for DPM. Incremental BHC (IBHC, [14]) is an extension of BHC to (dependent) NRMM models. Like BHC is a deterministic posterior inference algorithm for DPM, IBHC serves as a deterministic posterior inference algorithms for NRMM models. Unlike the original BHC that greedily builds trees, IBHC sequentially insert data points into trees, yielding scalable algorithm that is well suited for online inference. We first explain the generative model of trees, and then explain the sequential algorithm of IBHC.

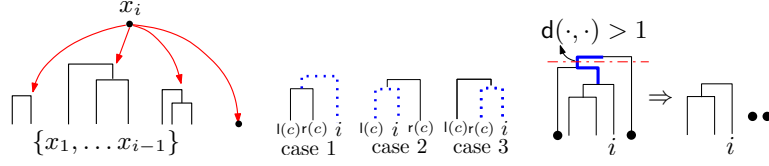

Figure 1: (Left) in IBHC, a new data point is inserted into one of the trees, or create a novel tree. (Middle) three possible cases in `SeqInsert`. (Right) after the insertion, the potential funcitons for the nodes in the blue bold path should be updated. If a updated $d(\cdot, \cdot) > 1$, the tree is split at that level.

IBHC aims to maximize the joint probability of the data $X$ and the auxiliary variable $u$:

$$P(dX, du) = \frac{u^{n-1}e^{-\psi_\rho(u)}du}{\Gamma(n)} \sum_{\Pi_{[n]}} \prod_{c \in \Pi_{[n]}} \kappa_\rho(|c|, u)P(dX_c) \qquad (7)$$

Let $t_c$ be a binary tree whose leaf nodes consist of the indices in $c$. Let $l(c)$ and $r(c)$ denote the left and right child of the set $c$ in tree, and thus the corresponding trees are denoted by $t_{l(c)}$ and $t_{r(c)}$. The generative probability of trees is described with the *potential function* [14], which is the unnormalized reformulation of the original definition [15]. The potential function of the data $X_c$ given the tree $t_c$ is recursively defined as follows:

$$\phi(X_c|h_c) := \kappa_\rho(|c|, u)P(dX_c), \quad \phi(X_c|t_c) = \phi(X_c|h_c) + \phi(X_{l(c)}|t_{l(c)})\phi(X_{r(c)}|t_{r(c)}). \qquad (8)$$

Here, $h_c$ is the hypothesis that $X_c$ was generated from a single cluster. The first therm $\phi(X_c|h_c)$ is proportional to the probability that $h_c$ is true, and came from the term inside the product of (7). The second term is proportional to the probability that $X_c$ was generated from more than two clusters embedded in the subtrees $t_{l(c)}$ and $t_{r(c)}$. The posterior probability of $h_c$ is then computed as

$$P(h_c|X_c, t_c) = \frac{1}{1 + d(l(c), r(c))}, \quad \text{where } d(l(c), r(c)) := \frac{\phi(X_{l(c)}|t_{l(c)})\phi(X_{r(c)}|t_{r(c)})}{\phi(X_c|h_c)}. \qquad (9)$$

$d(\cdot, \cdot)$ is defined to be the dissimilarity between $l(c)$ and $r(c)$. In the greedy construction, the pair of nodes with smallest $d(\cdot, \cdot)$ are merged at each iteration. When the minimum dissimilarity exceeds one ($P(h_c|X_c, t_c) < 0.5$), $h_c$ is concluded to be false and the construction stops. This is an important mechanism of BHC (and IBHC) that naturally selects the proper number of clusters. In the perspective of the posterior inference, this stopping corresponds to selecting the MAP partition that maximizes $P(\Pi_{[n]}|X, u)$. If the tree is built and the potential function is computed for the entire dataset $X$, a lower bound on the joint likelihood (7) is obtained [15, 14]:

$$\frac{u^{n-1}e^{-\psi_\rho(u)}du}{\Gamma(n)}\phi(X|t_{[n]}) \le P(dX, du). \qquad (10)$$

Now we explain the sequential tree construction of IBHC. IBHC constructs a tree in an incremental manner by inserting a new data point into an appropriate position of the existing tree, without computing dissimilarities between every pair of nodes. The procedure, which comprises three steps, is elucidated in Fig. 1.

**Step 1 (left):** Given $\{x_1, \ldots, x_{i-1}\}$, suppose that trees are built by IBHC, yielding to a partition $\Pi_{[i-1]}$. When a new data point $x_i$ arrives, this step assigns $x_i$ to a tree $t_{\hat{c}}$, which has the smallest distance, i.e., $\hat{c} = \arg\min_{c \in \Pi_{[i-1]}} d(i, c)$, or create a new tree $t_i$ if $d(i, \hat{c}) > 1$.

**Step 2 (middle):** Suppose that the tree chosen in Step 1 is $t_c$. Then Step 2 determines an appropriate position of $x_i$ when it is inserted into the tree $t_c$, and this is done by the procedure `SeqInsert`$(c, i)$. `SeqInsert`$(c, i)$ chooses the position of $i$ among three cases (Fig. 1). Case 1 elucidates an option where $x_i$ is placed on the top of the tree $t_c$. Case 2 and 3 show options where $x_i$ is added as a sibling of the subtree $t_{l(c)}$ or $t_{r(c)}$, respectively. Among these three cases, the one with the highest potential function $\phi(X_{c \cup i}|t_{c \cup i})$ is selected, which can easily be done by comparing $d(l(c), r(c))$, $d(l(c), i)$ and $d(r(c), i)$ [14]. If $d(l(c), r(c))$ is the smallest, then Case 1 is selected and the insertion terminates. Otherwise, if $d(l(c), i)$ is the smallest, $x_i$ is inserted into $t_{l(c)}$ and `SeqInsert`$(l(c), i)$ is recursively executed. The same procedure is applied to the case where $d(r(c), i)$ is smallest.

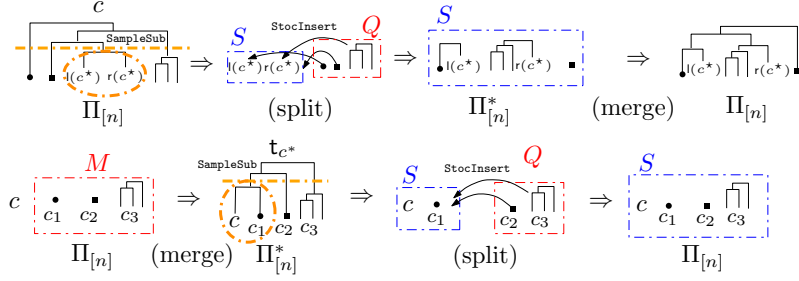

Figure 2: Global moves of tgMCMC. Top row explains the way of proposing split partition $\Pi_{[n]}^*$ from partition $\Pi_{[n]}$, and explains the way to retain $\Pi_{[n]}$ from $\Pi_{[n]}^*$. Bottom row shows the same things for merge case.

**Step 3 (right):** After Step 1 and 2 are applied, the potential functions of $t_{c \cup i}$ should be computed again, starting from the subtree of $t_c$ to which $x_i$ is inserted, to the root $t_{c \cup i}$. During this procedure, updated $\mathsf{d}(\cdot, \cdot)$ values may exceed 1. In such a case, we split the tree at the level where $\mathsf{d}(\cdot, \cdot) > 1$, and re-insert all the split nodes.

After having inserted all the data points in $X$, the auxiliary variable $u$ and hyperparameters for $\rho(dw)$ are resampled, and the tree is reconstructed. This procedure is repeated several times and the trees with the highest potential functions are chosen as an output.

## 3 Main results: A tree-guided MCMC procedure

IBHC should reconstruct trees from the ground whenever $u$ and hyperparameters are resampled, and this is obviously time consuming, and more importantly, converge is not guaranteed. Instead of completely reconstructing trees, we propose to refine the parts of existing trees with MCMC. Our algorithm, called tree-guided MCMC (tgMCMC), is a combination of deterministic tree-based inference and MCMC, where the trees constructed via IBHC guides MCMC to propose good-quality samples. tgMCMC initialize a chain with a single run of IBHC. Given a current partition $\Pi_{[n]}$ and trees $\{t_c \mid c \in \Pi_{[n]}\}$, tgMCMC proposes a novel partition $\Pi_{[n]}^*$ by global and local moves. Global moves split or merges clusters to propose $\Pi_{[n]}^*$, and local moves alters cluster assignments of individual data points via Gibbs sampling. We first explain the two key operations used to modify tree structures, and then explain global and local moves. More details on the algorithm can be found in the supplementary material.

### 3.1 Key operations

$\mathtt{SampleSub}(c, p)$: given a tree $t_c$, draw a subtree $t_{c'}$ with probability $\propto \mathsf{d}(\mathsf{l}(c'), \mathsf{r}(c')) + \epsilon$. $\epsilon$ is added for leaf nodes whose $\mathsf{d}(\cdot, \cdot) = 0$, and set to the maximum $\mathsf{d}(\cdot, \cdot)$ among all subtrees of $t_c$. The drawn subtree is likely to contain errors to be corrected by splitting. The probability of drawing $t_{c'}$ is multiplied to $p$, where $p$ is usually set to transition probabilities.

$\mathtt{StocInsert}(S, c, p)$: a stochastic version of IBHC. $c$ may be inserted to $c' \in S$ via $\mathtt{SeqInsert}(c', c)$ with probability $\frac{\mathsf{d}^{-1}(c', c)}{1 + \sum_{c' \in S} \mathsf{d}^{-1}(c', c)}$, or may just be put into $S$ (create a new cluster in $S$) with probability $\frac{1}{1 + \sum_{c' \in S} \mathsf{d}^{-1}(c', c)}$. If $c$ is inserted via $\mathtt{SeqInsert}$, the potential functions are updated accordingly, but the trees are *not* split even if the update dissimilarities exceed 1. As in $\mathtt{SampleSub}$, the probability is multiplied to $p$.

### 3.2 Global moves

The global moves of tgMCMC are tree-guided analogy to split-merge sampling. In split-merge sampling, a pair of data points are randomly selected, and split partition is proposed if they belong to the same cluster, or merged partition is proposed otherwise. Instead, tgMCMC finds the clusters that are highly likely to be split or merged using the dissimilarities between trees, which goes as follows in detail. First, we randomly pick a tree $t_c$ in uniform. Then, we compute $\mathsf{d}(c, c')$ for

$c' \in \Pi_{[n]} \backslash c$, and put $c'$ in a set $M$ with probability $(1 + \mathsf{d}(c, c'))^{-1}$ (the probability of merging $c$ and $c'$). The transition probability $q(\Pi_{[n]}^* | \Pi_{[n]})$ up to this step is $\frac{1}{|\Pi_{[n]}|} \prod_{c'} \frac{\mathsf{d}(c,c')^{\mathbb{1}[c' \notin M]}}{1 + \mathsf{d}(c,c')}$. The set $M$ contains candidate clusters to merge with $c$. If $M$ is empty, which means that there are no candidates to merge with $c$, we propose $\Pi_{[n]}^*$ by splitting $c$. Otherwise, we propose $\Pi_{[n]}^*$ by merging $c$ and clusters in $M$.

**Split case**: we start splitting by drawing a subtree $\mathsf{t}_{c^\star}$ by $\mathtt{SampleSub}(c, q(\Pi_{[n]}^* | \Pi_{[n]}))$ [1]. Then we split $c^\star$ to $S = \{\mathsf{l}(c^\star), \mathsf{r}(c^\star)\}$, destroy all the parents of $\mathsf{t}_{c^\star}$ and collect the split trees into a set $Q$ (Fig. 2, top). Then we reconstruct the tree by $\mathtt{StocInsert}(S, c', q(\Pi_{[n]}^* | \Pi_{[n]}))$ for all $c' \in Q$. After the reconstruction, $S$ has at least two clusters since we split $S = \{\mathsf{l}(c^\star), \mathsf{r}(c^\star)\}$ before insertion. The split partition to propose is $\Pi_{[n]}^* = (\Pi_{[n]} \backslash c) \cup S$. The reverse transition probability $q(\Pi_{[n]} | \Pi_{[n]}^*)$ is computed as follows. To obtain $\Pi_{[n]}$ from $\Pi_{[n]}^*$, we must merge the clusters in $S$ to $c$. For this, we should pick a cluster $c' \in S$, and put other clusters in $S \backslash c$ into $M$. Since we can pick any $c'$ at first, the reverse transition probability is computed as a sum of all those possibilities:

$$q(\Pi_{[n]} | \Pi_{[n]}^*) = \sum_{c' \in S} \frac{1}{|\Pi_{[n]}^*|} \prod_{c'' \in \Pi_{[n]}^* \backslash c'} \frac{\mathsf{d}(c', c'')^{\mathbb{1}[c'' \notin S]}}{1 + \mathsf{d}(c', c'')}, \tag{11}$$

**Merge case**: suppose that we have $M = \{c_1, \ldots, c_m\}$ [2]. The merged partition to propose is given as $\Pi_{[n]}^* = (\Pi_{[n]} \backslash M) \cup c_{m+1}$, where $c_{m+1} = \bigcup_{i=1}^m c_m$. We construct the corresponding binary tree as a cascading tree, where we put $c_1, \ldots c_m$ on top of $c$ in order (Fig. 2, bottom). To compute the reverse transition probability $q(\Pi_{[n]} | \Pi_{[n]}^*)$, we should compute the probability of splitting $c_{m+1}$ back into $c_1, \ldots, c_m$. For this, we should first choose $c_{m+1}$ and put nothing into the set $M$ to provoke splitting. $q(\Pi_{[n]} | \Pi_{[n]}^*)$ up to this step is $\frac{1}{|\Pi_{[n]}^*|} \prod_{c'} \frac{\mathsf{d}(c_{m+1}, c')}{1 + \mathsf{d}(c_{m+1}, c')}$. Then, we should sample the parent of $c$ (the subtree connecting $c$ and $c_1$) via $\mathtt{SampleSub}(c_{m+1}, q(\Pi_{[n]} | \Pi_{[n]}^*))$, and this would result in $S = \{c, c_1\}$ and $Q = \{c_2, \ldots, c_m\}$. Finally, we insert $c_i \in Q$ into $S$ via $\mathtt{StocInsert}(S, c_i, q(\Pi_{[n]} | \Pi_{[n]}^*))$ for $i = 2, \ldots, m$, where we select each $c^{(i)}$ to create a new cluster in $S$. Corresponding update to $q(\Pi_{[n]} | \Pi_{[n]}^*)$ by $\mathtt{StocInsert}$ is,

$$q(\Pi_{[n]} | \Pi_{[n]}^*) \leftarrow q(\Pi_{[n]} | \Pi_{[n]}^*) \cdot \prod_{i=2}^m \frac{1}{1 + \mathsf{d}^{-1}(c, c_i) + \sum_{j=1}^{i-1} \mathsf{d}^{-1}(c_j, c_i)}. \tag{12}$$

Once we've proposed $\Pi_{[n]}^*$ and computed both $q(\Pi_{[n]}^* | \Pi_{[n]})$ and $q(\Pi_{[n]} | \Pi_{[n]}^*)$, $\Pi_{[n]}^*$ is accepted with probability $\min\{1, r\}$ where $r = \frac{p(dX, du, \Pi_{[n]}*)q(\Pi_{[n]} | \Pi_{[n]}^*)}{p(dX, du, \Pi_{[n]})q(\Pi_{[n]}^* | \Pi_{[n]})}$.

**Ergodicity of the global moves**: to show that the global moves are ergodic, it is enough to show that we can move an arbitrary point $i$ from its current cluster $c$ to any other cluster $c'$ in finite step. This can easily be done by a single split and merge moves, so the global moves are ergodic.

**Time complexity of the global moves**: the time complexity of $\mathtt{StocInsert}(S, c, p)$ is $O(|S| + h)$, where $h$ is a height of the tree to insert $c$. The total time complexity of split proposal is mainly determined by the time to execute $\mathtt{StocInsert}(S, c, p)$. This procedure is usually efficient, especially when the trees are well balanced. The time complexity to propose merged partition is $O(|\Pi_{[n]}| + M)$.

### 3.3 Local moves

In local moves, we resample cluster assignments of individual data points via Gibbs sampling. If a leaf node $i$ is moved from $c$ to $c'$, we detach $i$ from $\mathsf{t}_c$ and run $\mathtt{SeqInsert}(c', i)$ [3]. Here, instead of running Gibbs sampling for all data points, we run Gibbs sampling for a subset of data points $S$, which is formed as follows. For each $c \in \Pi_{[n]}$, we draw a subtree $\mathsf{t}_{c'}$ by $\mathtt{SampleSub}$. Then, we

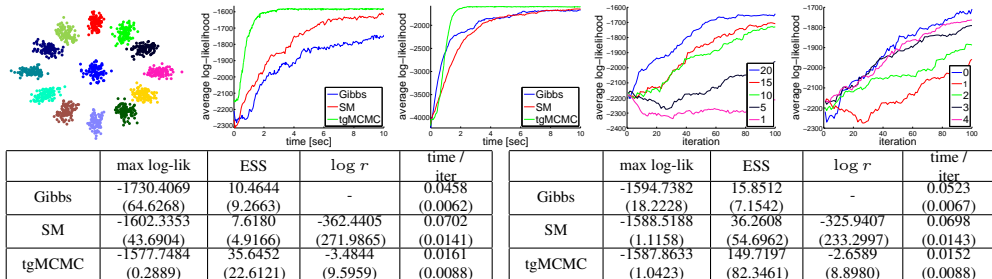

| | max log-lik | ESS | log $r$ | time / iter | | max log-lik | ESS | log $r$ | time / iter |
|---|---|---|---|---|---|---|---|---|---|
| Gibbs | -1730.4069 (64.6268) | 10.4644 (9.2663) | - | 0.0458 (0.0062) | Gibbs | -1594.7382 (18.2228) | 15.8512 (7.1542) | - | 0.0523 (0.0067) |
| SM | -1602.3353 (43.6904) | 7.6180 (4.9166) | -362.4405 (271.9865) | 0.0702 (0.0141) | SM | -1588.5188 (1.1158) | 36.2608 (54.6962) | -325.9407 (233.2997) | 0.0698 (0.0143) |
| tgMCMC | -1577.7484 (0.2889) | 35.6452 (22.6121) | -3.4844 (9.5959) | 0.0161 (0.0088) | tgMCMC | -1587.8633 (1.0423) | 149.7197 (82.3461) | -2.6589 (8.8980) | 0.0152 (0.0088) |

Figure 3: Experimental results on toy dataset. (Top row) scatter plot of toy dataset, log-likelihoods of three samplers with DP, log-likelihoods with NGGP, log-likelihoods of tgMCMC with varying $G$ and varying $D$. (Bottom row) The statistics of three samplers with DP and the statistics of three samplers with NGGP.

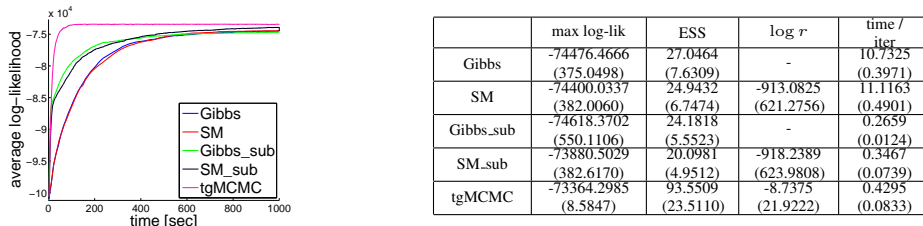

| | max log-lik | ESS | log $r$ | time / iter |
|---|---|---|---|---|
| Gibbs | -74476.4666 (375.0498) | 27.0464 (7.6309) | - | 10.7325 (0.3971) |
| SM | -74400.0337 (382.0060) | 24.9432 (6.7474) | -913.0825 (621.2756) | 11.1163 (0.4901) |
| Gibbs_sub | -74618.3702 (550.1106) | 24.1818 (5.5523) | - | 0.2659 (0.0124) |
| SM_sub | -73880.5029 (382.6170) | 20.0981 (4.9512) | -918.2389 (623.9808) | 0.3467 (0.0739) |
| tgMCMC | -73364.2985 (8.5847) | 93.5509 (23.5110) | -8.7375 (21.9222) | 0.4295 (0.0833) |

Figure 4: Average log-likelihood plot and the statistics of the samplers for 10K dataset.

draw a subtree of $t_{c'}$ again by `SampleSub`. We repeat this subsampling for $D$ times, and put the leaf nodes of the final subtree into $S$. Smaller $D$ would result in more data points to resample, so we can control the tradeoff between iteration time and mixing rates.

**Cycling**: at each iteration of tgMCMC, we cycle the global moves and local moves, as in split-merge sampling. We first run the global moves for $G$ times, and run a single sweep of local moves. Setting $G = 20$ and $D = 2$ were the moderate choice for all data we've tested.

## 4  Experiments

In this section, we compare marginal Gibbs sampler (Gibbs), split-merge sampler (SM) and tgMCMC on synthetic and real datasets.

### 4.1  Toy dataset

We first compared the samplers on simple toy dataset that has 1,300 two-dimensional points with 13 clusters, sampled from the mixture of Gaussians with predefined means and covariances. Since the partition found by IBHC is almost perfect for this simple data, instead of initializing with IBHC, we initialized the binary tree (and partition) as follows. As in IBHC, we sequentially inserted data points into existing trees with a random order. However, instead of inserting them via `SeqInsert`, we just put data points on top of existing trees, so that no splitting would occur. tgMCMC was initialized with the tree constructed from this procedure, and Gibbs and SM were initialized with corresponding partition. We assumed the Gaussian-likelihood and Gaussian-Wishart base measure,

$$H(d\mu, d\Lambda) = \mathcal{N}(d\mu|m, (r\Lambda)^{-1})\mathcal{W}(d\Lambda|\Psi^{-1}, \nu), \tag{13}$$

where $r = 0.1$, $\nu = d+6$, $d$ is the dimensionality, $m$ is the sample mean and $\Psi = \Sigma/(10 \cdot \det(\Sigma))^{1/d}$ ($\Sigma$ is the sample covariance). We compared the samplers using both DP and NGGP priors. For tgMCMC, we fixed the number of global moves $G = 20$ and the parameter for local moves $D = 2$, except for the cases where we controlled them explicitly. All the samplers were run for 10 seconds, and repeated 10 times. We compared the joint log-likelihood $\log p(dX, \Pi_{[n]}, du)$ of samples and the effective sample size (ESS) of the number of clusters found. For SM and tgMCMC, we compared the average log value of the acceptance ratio $r$. The results are summarized in Fig. 3. As shown in

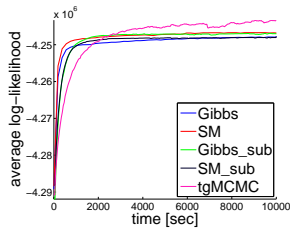

| | max log-lik | ESS | $\log r$ | time / iter |
|---|---|---|---|---|
| Gibbs | -4247895.5166 (1527.0131) | 18.1758 (7.2028) | - | 186.2020 (2.9030) |
| SM | -4246689.8072 (1656.2299) | 28.6608 (18.1896) | -3290.2988 (2617.6750) | 186.9424 (2.2014) |
| Gibbs_sub | -4246878.3344 (1391.1707) | 13.8057 (4.5723) | - | 49.7875 (0.9400) |
| SM_sub | -4248034.0748 (1703.6653) | 18.5764 (18.6368) | -3488.9523 (3145.9786) | 49.9563 (0.8667) |
| tgMCMC | -4243009.3500 (1101.0383) | 3.1274 (2.6610) | -256.4831 (218.8061) | 42.4176 (2.0534) |

Figure 5: Average log-likelihood plot and the statistics of the samplers for NIPS corpus.

the log-likelihood trace plot, tgMCMC quickly converged to the ground truth solution for both DP and NGGP cases. Also, tgMCMC mixed better than other two samplers in terms of ESS. Comparing the average $\log r$ values of SM and tgMCMC, we can see that the partitions proposed by tgMCMC is more often accepted. We also controlled the parameter $G$ and $D$; as expected, higher $G$ resulted in faster convergence. However, smaller $D$ (more data points involved in local moves) did not necessarily mean faster convergence.

## 4.2 Large-scale synthetic dataset

We also compared the three samplers on larger dataset containing 10,000 points, which we will call as 10K dataset, generated from six-dimensional mixture of Gaussians with labels drawn from $PY(3, 0.8)$. We used the same base measure and initialization with those of the toy datasets, and used the NGGP prior, We ran the samplers for 1,000 seconds and repeated 10 times. Gibbs and SM were too slow, so the number of samples produced in 1,000 seconds were too small. Hence, we also compared Gibbs_sub and SM_sub, where we uniformly sampled the subset of data points and ran Gibbs sweep only for those sampled points. We controlled the subset size to make their running time similar to that of tgMCMC. The results are summarized in Fig. 4. Again, tgMCMC outperformed other samplers both in terms of the log-likelihoods and ESS. Interestingly, SM was even worse than Gibbs, since most of the samples proposed by split or merge proposal were rejected. Gibbs_sub and SM_sub were better than Gibbs and SM, but still failed to reach the best state found by tgMCMC.

## 4.3 NIPS corpus

We also compared the samplers on NIPS corpus[4], containing 1,500 documents with 12,419 words. We used the multinomial likelihood and symmetric Dirichlet base measure $\mathrm{Dir}(0.1)$, used NGGP prior, and initialized the samplers with normal IBHC. As for the 10K dataset, we compared Gibbs_sub and SM_sub along. We ran the samplers for 10,000 seconds and repeated 10 times. The results are summarized in Fig. 5. tgMCMC outperformed other samplers in terms of the log-likelihood; all the other samplers were trapped in local optima and failed to reach the states found by tgMCMC. However, ESS for tgMCMC were the lowest, meaning the poor mixing rates. We still argue that tgMCMC is a better option for this dataset, since we think that finding the better log-likelihood states is more important than mixing rates.

## 5 Conclusion

In this paper we have presented a novel inference algorithm for NRMM models. Our sampler, called tgMCMC, utilized the binary trees constructed by IBHC to propose good quality samples. tgMCMC explored the space of partitions via global and local moves which were guided by the potential functions of trees. tgMCMC was demonstrated to be outperform existing samplers in both synthetic and real world datasets.

**Acknowledgments**: This work was supported by the IT R&D Program of MSIP/IITP (B0101-15-0307, Machine Learning Center), National Research Foundation (NRF) of Korea (NRF-2013R1A2A2A01067464), and IITP-MSRA Creative ICT/SW Research Project.

## Footnotes

[1]Here, we restrict $\mathtt{SampleSub}$ to sample non-leaf nodes, since leaf nodes cannot be split.

[2]We assume that clusters are given their own indices (such as hash values) so that they can be ordered.

[3]We *do not* split even if the update dissimilarity exceed one, as in $\mathtt{StocInsert}$.

[4]https://archive.ics.uci.edu/ml/datasets/Bag+of+Words

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
