[Supplementary Material · supple.pdf]

# Supplementary material for Tree-Guided MCMC Inference for Normalized Random Measure Mixture Models

**Juho Lee**
Department of Computer Science and Engineering
Pohang University of Science and Technology
77 Cheongam-ro, Nam-gu, Pohang 790-784, Korea
stonecold@postech.ac.kr

**Seungjin Choi**
Department of Computer Science and Engineering
Pohang University of Science and Technology
77 Cheongam-ro, Nam-gu, Pohang 790-784, Korea
seungjin@postech.ac.kr

In this supplementary material, we explain our proposing algorithm in more detail, with illustrative examples and figures. All the notations follow those of our papers.

## 1 Key operations

We first explain two key operations used for tgMCMC.

$\texttt{SampleSub}(c, p)$: given a tree $\mathsf{t}_c$, $\texttt{SampleSub}(c, p)$ aims to draw a subtree $\mathsf{t}_{c'}$ of $\mathsf{t}_c$. We want $\mathsf{t}_{c'}$ to have high dissimilarity between its subtrees, so we draw $\mathsf{t}_{c'}$ with probability proportional to $\mathsf{d}(\mathsf{l}(c'), \mathsf{r}(c')) + \epsilon$, where $\epsilon$ is the maximum dissimilarity between subtrees. $\epsilon$ is added for leaf nodes, whose dissimilarity between subtrees are defined to be zero. See Figure 1.

$$\epsilon = \max_i \mathsf{d}(\mathsf{l}(c_i), \mathsf{r}(c_i))$$

$$p_i = \frac{\mathsf{d}(\mathsf{l}(c_i), \mathsf{r}(c_i)) + \epsilon}{\sum_{j=1}^{7} \{\mathsf{d}(\mathsf{l}(c_j), \mathsf{r}(c_j)) + \epsilon\}}$$

Figure 1: An illustrative example for $\texttt{SampleSub}$ operation.

As depicted in Figure 1, the probability of drawing the subtree $\mathsf{t}_{c_i}$ is computed as $p_i$. If we draw $\mathsf{t}_{c_i}$ as a result, corresponding probability $p_i$ is multiplied into $p$; $p \leftarrow p \cdot p_i$. This procedure is needed to compute the forward transition probability. If $\texttt{SampleSub}$ takes another argument as $\texttt{SampleSub}(c, c', p)$, we don't do the actual sampling but compute the probability of drawing subtree $\mathsf{t}_{c'}$ from $\mathsf{t}_c$ and multiply the probability to $p$. In above example, $\texttt{SampleSub}(c, c_i, p)$ does $p \leftarrow p \cdot p_i$. This operation is needed to compute the reverse transition probability.

$\texttt{StocInsert}(S, c, p)$: given a set of clusters $S$ (and corresponding trees), $\texttt{StocInsert}$ inserts $\mathsf{t}_c$ into one of the trees $\mathsf{t}_{c'}(c' \in S)$, or just put $c$ in $S$ without insertion. If $\mathsf{t}_c$ is inserted into $\mathsf{t}_{c'}$, the insertion is done by $\texttt{SeqInsert}(c, c')$. See Figure 2 for example. The set $S$ contains three clusters, and $\mathsf{t}_c$ may be inserted into $c_i \in S$ via $\texttt{SeqInsert}(c_i, c)$ with probability $p_i$, or $c$ may just be put into $S$ without insertion with probability $p^*$. In the original IBHC algorithm, $\mathsf{t}_c$ is inserted into the tree $c_i$ with smallest $\mathsf{d}(c, c_i)$, or put into $S$ without insertion if the minimum $\mathsf{d}(c, c_i)$ exceeds one. $\texttt{StocInsert}$ is a probabilistic operation of this procedure, where the probability $p_i$

is proportional to $\mathsf{d}^{-1}(c, c_i)$. As emphasized in the paper, the trees are not split after the update of the dissimilarities of inserted trees; this is to ensure reversibility and make the computation of

$$p^* = \frac{1}{1+\sum_{i=1}^3 \mathsf{d}^{-1}(c_i,c)}$$

$$p_i = \frac{\mathsf{d}^{-1}(c_i,c)}{1+\sum_{i=1}^3 \mathsf{d}^{-1}(c_i,c)}$$

$$S = \{c_1, c_2, c_3\}$$

Figure 2: An illustrative example for `StocInsert` operation.

transition probability easier. Even if some dissimilarity exceeds one after the insertion, we expect `StocInsert` operation to find those nodes needed to be split. After the insertion, corresponding probability is multiplied to $p$, as in `StocInsert` operation. If `StocInsert` takes another arguments as $\mathtt{StocInsert}(S, c, c', p)$, we don't do insertion but compute the probability of inserting $c$ into $c' \in S$ and multiply it to $p$. In above example, $\mathtt{StocInsert}(S, c, c_i, p)$ multiplies $p \leftarrow p \cdot p_i$. $\mathtt{StocInsert}(S, c, \varnothing, p)$ multiplies the probability of creating a new tree with $c$, $p \leftarrow p \cdot p^*$.

## 2 Global moves

In this section, we explain global moves for our tgMCMC.

### 2.1 Selection between splitting and merging

In global moves, we randomly choose between splitting and merging operations. The basic principle is this; pick a tree, and find candidate trees that can be merged with the picked tree. If there is any candidate, invoke merge operation, and invoke split operation otherwise.

As an example, suppose that our current partition is $\Pi_{[n]} = \{c_i\}_{i=1}^6$, with 6 clusters. We first uniformly pick a cluster among them, and initialize the forward transition probability $q(\Pi_{[n]}^*|\Pi_{[n]}) = 1/6$. Suppose that $c_1$ is selected. Then, for $\{c_i\}_{i=2}^5$, we put $c_i$ into a set $M$ with probability $(1 + \mathsf{d}(c_1, c_i))^{-1}$; note that this probability is equal to $p(\mathsf{h}_{c_1 \cup c_i}|X_{c_1 \cup c_i})$, the probability of merging $c_1$ and $c_i$. The splitting is invoked if $M$ contains no cluster, and merging is invoked otherwise.

### 2.2 Splitting

Suppose that $M$ contains no cluster. The forward transition probability up to this is

$$q(\Pi_{[n]}^*|\Pi_{[n]}) = \frac{1}{6}\prod_{i=2}^6 \frac{\mathsf{d}(c_1, c_i)}{1 + \mathsf{d}(c_1, c_i)}. \tag{1}$$

Now suppose that $c_1$ looks like in Figure 3. The split proposal $\Pi_{[n]}^*$ is then proposed according to the following procedure. First, a subtree $c^\star$ of $c_1$ is sampled via `SampleSub` procedure. Then,

$$S = \{\mathsf{l}(c^\star), \mathsf{r}(c^\star)\}$$
$$Q = \{c_7, c_8\}$$

$\mathtt{StocInsert}(S, c_7, q(\Pi_{[n]}^*|\Pi_{[n]}))$
$\mathtt{StocInsert}(S, c_8, q(\Pi_{[n]}^*|\Pi_{[n]}))$

$\mathtt{SampleSub}(c_1, q(\Pi_{[n]}^*|\Pi_{[n]}))$

$$\Pi_{[n]}^* = \{c_i\}_{i=2}^6 \cup \{c_9, c_{10}\}.$$

Figure 3: An example for proposing split partition.

the tree is cut at $c^\star$, collecting $S = \{\mathsf{l}(c^\star), \mathsf{r}(c_1^\star)\}$ and remaining split nodes $Q = \{c_7, c_8\}$. Nodes in $Q$ are inserted into $S$ via `StocInsert`. Since $(\mathsf{l}(c^\star), \mathsf{r}(c_1^\star))$ were initialized to be split in $S$, the resulting partition must have more than two clusters. During `SampleSub` and `StocInsert`, every intermediate transition probabilities are multiplied into $q(\Pi_{[n]}^*|\Pi_{[n]})$. The final partition $\Pi_{[n]}^*$ to propose is $\{c^{(i)}\}_{i=2}^6 \cup \{c_9, c_{10}\}$.

$$c_9$$
$$c_8$$
$$c_7$$
$$c_1 \quad c_2 \quad c_3 \quad c_4$$
$$\texttt{SampleSub}(c_9, c_7, q(\Pi_{[n]}|\Pi_{[n]}^*))$$

$$\Rightarrow \quad S = \{c_1, c_2\}$$
$$Q = \{c_3, c_4\}$$
$$\texttt{StocInsert}(S, c_3, \varnothing, q(\Pi_{[n]}|\Pi_{[n]}^*))$$
$$\texttt{StocInsert}(S, c_4, \varnothing, q(\Pi_{[n]}|\Pi_{[n]}^*))$$

$$\Rightarrow \quad c_1 \quad c_2 \quad c_3 \quad c_4$$

Figure 4: An example for proposing merged partition, and how to compute the reverse transition probability in that case.

## 2.3 Merging

Suppose that $M$ contains three clusters; $M = \{c_2, c_3, c_4\}$. The forward transition probability up to this is

$$q(\Pi_{[n]}^*|\Pi_{[n]}) = \frac{1}{6} \prod_{i=2}^{4} \frac{1}{1 + \mathsf{d}(c_1, c_i)} \prod_{i=5}^{6} \frac{\mathsf{d}(c_1, c_i)}{1 + \mathsf{d}(c_1, c_i)}. \tag{2}$$

In partition space, there is only one way to merge $c_1$ and $M = \{c_2, c_3, c_4\}$ into a single cluster $c_9 = c_1 \cup c_2 \cup c_3 \cup c_4$, so no transition probability is multiplied. However, there can be many trees representing $c_9$, since we can merge the nodes in any order or topology, in terms of tree. Hence, we fix the merged tree to be a cascaded tree, merged in order of indices of nodes; see Figure 4. As we wrote in our paper, we assume that each nodes are given unique indices (such as hash values) so that they can be sorted in order. In this example, the indices for $c_1$ and $c_2$ are 1 and 2, respectively. Then, we build a cascaded tree, merged in order $c_1, c_2, c_3, c_4$. The resulting merged partition is then $\Pi_{[n]}^* = \{c_5, c_6, c_9\}$.

## 2.4 Computing reverse transition probability for splitting

Now we explain how to compute the reverse transition probability $q(\Pi_{[n]}|\Pi_{[n]}^*)$ for splitting case. We start from the illustrative partition $\Pi_{[n]}^*$ in subsection 2.2. Starting from $\Pi_{[n]}^*$, we must merge $c_9$ and $c_{10}$ back into $c_1 = c_9 \cup c_{10}$ to retrieve $\Pi_{[n]}$. For this, there are two possibilities: we first pick $c_9$ and collect $M = \{c_{10}\}$, or pick $c_{10}$ and collect $M = \{c_9\}$. Hence, the reverse transition probability is computed as

$$
\begin{aligned}
q(\Pi_{[n]}|\Pi_{[n]}^*) &= \frac{1}{7}\left( \sum_{i=1}^{5} \frac{\mathsf{d}(c_9, c_i)}{1 + \mathsf{d}(c_9, c_i)} + \frac{1}{1 + \mathsf{d}(c_9, c_{10})} \right) \\
&+ \frac{1}{7}\left( \sum_{i=1}^{5} \frac{\mathsf{d}(c_{10}, c_i)}{1 + \mathsf{d}(c_{10}, c_i)} + \frac{1}{1 + \mathsf{d}(c_{10}, c_9)} \right).
\end{aligned}
\tag{3}
$$

As we explained in subsection 2.3, no more reverse transition probabilities are needed since there is only one way to merge nodes back into $c_1$ in partition space.

## 2.5 Computing reverse transition probability for merging

We explain how to compute the reverse transition probability for merging case, starting from the illustrative partition in subsection 2.3, $\Pi_{[n]}^* = \{c_5, c_6, c_9\}$. See Figure 4. To retrieve $\Pi_{[n]} = \{c_i\}_{i=1}^{6}$, we should split $c_9$ into $c_1, c_2, c_3, c_4$. For this, we should first invoke split operation for $c_9$; pick $c_9$ and collect $M = \varnothing$. The reverse transition probability up to this is

$$q(\Pi_{[n]}|\Pi_{[n]}^*) = \frac{1}{3}\left( \frac{\mathsf{d}(c_9, c_5)}{1 + \mathsf{d}(c_9, c_5)} + \frac{\mathsf{d}(c_9, c_6)}{1 + \mathsf{d}(c_9, c_6)} \right). \tag{4}$$

Now, we should split $c_9$ to retrieve the original partition. The achieve this, we should first pick direct parent of $c_1$ and $c_2$ ($c_7$ in Figure 4) via $\texttt{SampleSub}$. The reverse transition probability for this is computed by $\texttt{SampleSub}(c_9, c_7, q(\Pi_{[n]}|\Pi_{[n]}^*))$, and we have $S = \{c_1, c_2\}$ and $Q = \{c_3, c_4\}$. To get the original partition, $c_3$ and $c_4$ should be put into $S$ without insertion, and the reverse transition probability for this is computed by $\texttt{StocInsert}(S, c_3, \varnothing, q(\Pi_{[n]}|\Pi_{[n]}^*))$ and $\texttt{StocInsert}(S, c_4, \varnothing, q(\Pi_{[n]}|\Pi_{[n]}^*))$.

# 3  Local moves

Local moves resample cluster assignments of individual data points (leaf nodes) with Gibbs sampling. However, instead of running Gibbs sampling for full data, we select a random subset $S$ and run Gibbs sampling for data points in $S$. $S$ is constructed as follows. Given a partition $\Pi_{[n]}$, for each $c \in \Pi_{[n]}$, we sample its subtree via `SampleSub`. Then we sample a subtree of drawn subtree again with `SampleSub`. This procedure is repeated for $D$ times, where $D$ is a parameter set by users. Figure 5 depicts the subsampling procedure for $c \in \Pi_{[n]}$ with $D = 2$. As a result of Figure 5, the

$$\text{SampleSub}(c, p) \qquad \text{SampleSub}(c_1, p)$$

Figure 5: An example for local moves.

leaf nodes $\{i, j, k\}$ are added to $S$. After having collected $S$ for all $c \in \Pi_{[n]}$, we run Gibbs sampling for those leaf nodes in $S$, and if a leaf node $i$ should be moved from $c$ to another $c'$, we insert $i$ into $c'$ with `SeqInsert`$(c', i)$. We can control the tradeoff between mixing rate and speed by controlling $D$; higher $D$ means more elements in $S$, and thus more data points are moved by Gibbs sampling.