[Reviews · NeurIPS 2015]

Submitted by Assigned_Reviewer_1

The authors present an inference method for a class of cluster models involving normalized random measures.

The method is a hybrid between standard MCMC moves and a version of hierarchical agglomerative clustering that is shown to draw samples that quickly find areas of high probability mass in the posterior distribution.

It is unclear to me if this method will generate a markov chain with the true posterior as its target, or creates some approximation.

It appears that finding states with high marginal likelihood is the primary goal, and correctly proportioned exploration of the posterior not much of a concern.

Is that the case?

If so - maybe make that point clear.

Summary: This is a light review: this paper is clearly written and provides compelling empirical results.

The method, a hybrid of hierarchical agglomerative clustering and mcmc moves seem novel and effective.

Submitted by Assigned_Reviewer_2

The paper proposes an efficient MCMC inference for normalized random measure mixture models. Borrowing the idea of hierarchical Bayesian clustering,

the inference algorithm uses tree structures to represent cluster and develops stochastic tree operations to generate proposals; it inherits the benefit of incremental Bayesian clustering algorithm (ICBH)---the operation over trees is efficient and the tree-guided proposals are empirically effective; it overcomes the deficiency of IBHC by developing the inference rigorously in MCMC framework: the transition probability is clearly defined and the convergence is theoretically guaranteed.

The paper is well written and the work is solid. The only issue to me is that on real dataset, the proposed algorithm shows slower mixing rate than the competing methods, which is a little disappointing; maybe the authors should test more real datasets for confirmation.
Summary: The paper proposes a tree-guided MCMC inference algorithm for normalized random measure mixture models; it not only inherits the efficiency of tree operations and the effectiveness of tree-structured proposals, as in ICBH, but also rigorously guarantees convergence; it is a piece of solid work.

Author Feedback
Author rebuttal: We thank all the reviewers for their helpful and constructive comments. In this rebuttal, we try to answer some questions or concerns raised by reviewers. Before that, we want to thank for the reviewers to recognize the contribution of our proposing methods

Assigned_Reviewer_1 (R1)
We agree that our method is complicated and not easy to implement, so we are planning to release our source codes.

Assigned_Reviewer_4 (R4)
We agree that the low ESS of tgMCMC for NIPS corpus might be disappointing, but we want to emphasize that among all the samplers, only tgMCMC succeeded to reach higher likelihood state, and we carefully argue that reaching higher likelihood is more important than higher mixing rates.

Assigned_Reviewer_5 (R5)
Regarding a connection to split-merge sampling, at first glance our method tgMCMC appears to be a twist of split-merge sampler. But the truth is that it is NOT a twist! In fact, the split-merge behavior of tgMCMC came from IBHC, NOT from an extension of split-merge sampler. In contrast to the split-merge sampler, tgMCMC is able to split or merge more than two clusters at a time, which cannot be done in the split-merge sampler. This is because split or merge is guided by a tree (comprising the information on cluster structure) built by IBHC in tgMCMC, which clearly emphasizes that tgMCMC is NOT a twist to the split-merge sampler. Regarding our experiments, we agree that experiments on synthetic data are focused on simple mixture models. However, we believe that those experiments are sufficient to demonstrate the merits of our tgMCMC, since we also included experiments on large-scale real-world data.

Assigned_Reviewer_6 (R6)
As we stated earlier, we agree that the presentation should be improved, and we are going to release the source codes for our algorithm.

Assigned_Reviewer_7 (R7)
Our sampler converges to the true posterior since both global moves and local moves are ergodic moves, and the cycling of two ergodic chains are also ergodic. We will make this point clearer in the paper.